# We're Afraid Language Models Aren't Modeling Ambiguity

**Alisa Liu**[♡]  **Zhaofeng Wu**[♦]  **Julian Michael**[♠]  **Alane Suhr**[♣△]  **Peter West**[♡♣]
**Alexander Koller**[♣♥]  **Swabha Swayamdipta**[♢]  **Noah A. Smith**[♡♣]  **Yejin Choi**[♡♣]

[♡]Paul G. Allen School of Computer Science & Engineering, University of Washington
[♣]Allen Institute for AI   [♢]University of Southern California   [△]UC Berkeley
[♥]Saarland University   [♠]New York University   [♦]Massachusetts Institute of Technology
alisaliu@cs.washington.edu

## Abstract

Ambiguity is an intrinsic feature of natural language. Managing ambiguity is a key part of human language understanding, allowing us to anticipate misunderstanding as communicators and revise our interpretations as listeners. As language models are increasingly employed as dialogue interfaces and writing aids, handling ambiguous language is critical to their success. We capture ambiguity in a sentence through its effect on *entailment* relations with another sentence, and collect AMBIENT,[1] a linguist-annotated benchmark of 1,645 examples with diverse kinds of ambiguity. We design a suite of tests based on AMBIENT, presenting the first evaluation of pretrained LMs to recognize ambiguity and disentangle possible meanings. We find that the task remains extremely challenging, including for GPT-4, whose generated disambiguations are considered correct only 32% of the time in crowdworker evaluation, compared to 90% for disambiguations in our dataset. Finally, to illustrate the value of ambiguity-sensitive tools, we show that a multilabel NLI model can flag political claims *in the wild* that are misleading due to ambiguity. We encourage the field to rediscover the importance of ambiguity for NLP.

## 1 Introduction

> *Ambiguity seems to be an essential, indispensable element for the transfer of information from one place to another by words.* — Thomas (1974), as referenced in the epilogue of Grosz (1977)

Ambiguity is an intrinsic feature of language, allowing speakers to balance efficiency and clarity in communication (Zipf, 1949; Piantadosi et al., 2012). Language understanding thus requires recognizing the presence of multiple interpretations:

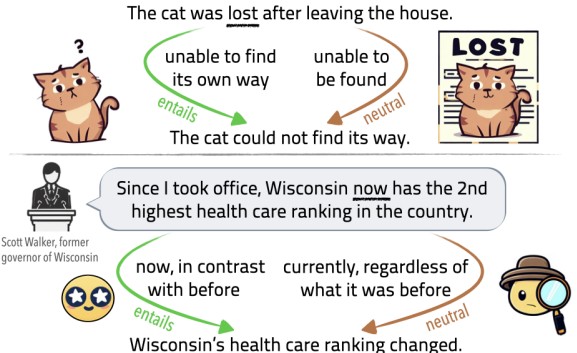

Figure 1: Ambiguity can be the result of innocent miscommunication (**top**), or deliberately used to mislead one's listeners (**bottom**). For instance, a cat may be lost in the sense of being confused about its whereabouts (entailment edge), or lost in the sense that *others* cannot find it (neutral edge). Each example in AMBIENT contains a *set* of labels corresponding to plausible readings, along with a disambiguating rewrite for each reading.

as communicators, we anticipate the possibility of misunderstanding; as listeners, we ask clarifying questions, disambiguate meanings on the basis of a wide range of contextual factors, and backtrack and revise our earlier interpretations as needed. Beyond unintended miscommunication, ambiguity is also an effective tool for sending covert messages, e.g., out of politeness or to mislead one's listeners while avoiding accountability (see Figure 1).

As language models (LMs) are increasingly employed to act as dialogue agents (OpenAI, 2022; Shuster et al., 2022) or to aid human communication as writing aids (Lee et al., 2022), being able to work with ambiguous language will make them more effective. This skill would support adaptation to different contexts, clearer communication, and identification of misleading or deceptive language. Yet, the ability of pretrained LMs to recognize ambiguity and *disentangle possible meanings* remains unstudied, partly because ambiguous instances are systematically excluded in the curation of benchmarks (Beigman Klebanov and Beigman, 2009).

---

[1]Data and code can be found at https://github.com/alisawuffles/ambient

We present **AMBIENT**, **Ambi**guity in **Ent**ailment, an English benchmark of 1,645 examples covering a variety of lexical, syntactic, and pragmatic ambiguities, and more broadly sentences which can be plausibly read as conveying one of multiple different messages. Formally characterizing ambiguity requires a choice of meaning representation to distinguish between possible interpretations, and enumerating the full set of interpretations can be tricky or impractical.[2] Thus, we adopt a *functional* approach: using the natural language inference (NLI) task format, we characterize ambiguity in the premise and/or hypothesis by its effect on entailment relations.[3]

Each AMBIENT example consists of a premise and hypothesis pair, assigned a *set* of labels (among entailment, neutral, and contradiction), along with *disambiguating rewrites* corresponding to each label when multiple are plausible (see Table 1 for examples). Examples are collected through two approaches: manual curation to target textbook ambiguities, and expert annotation of automatically generated unlabeled examples to uncover more diverse phenomena. Through analysis, we find that crowdworkers can reliably distinguish different readings of an ambiguous sentence and their impact on entailment choices; thus we can explicitly characterize the underlying reasons for uncertainty that would otherwise surface as "disagreement" (§3).

We design a suite of tests based on AMBIENT to investigate the extent to which understanding of ambiguity is acquired during pretraining of large LMs (§4). These tests evaluate whether LMs can directly produce relevant disambiguations, recognize possible interpretations, and model different interpretations in their continuation distributions. We find that these tasks remain extremely challenging, including for the recent GPT-4 (OpenAI, 2023).

Therefore, we additionally investigate whether LMs can be *finetuned* on existing NLI data for the less demanding task of ambiguity *recognition*, without explicit disambiguation (§5). We adapt several finetuned NLI models to a multilabel setting, and find that the best model predicts the exact label set in only 43.6% of instances, suggesting that the NLI task is much more challenging when formulated to account for ambiguity.

Finally, to illustrate the value of ambiguity-sensitive tools, we present a case study of how a multilabel NLI model can be used to detect misleading political claims *in the wild*. We find that the strongest model from §5, despite its limitations, can not only recover claims flagged by fact-checkers as ambiguous, but highlight previously unidentified ambiguous claims, indicating the promise of such tools to aid real-world communication.

The simplifying assumption that text has only one interpretation has facilitated the development of large-scale benchmarks, yet limits the depth of what these benchmarks can evaluate. In this work we show that sensitivity to ambiguity—a fundamental aspect of human language understanding—is lacking in our ever-larger models, and illustrate the value such understanding could bring.

## 2  AMBIENT

Traditionally, the NLI task requires predicting whether a premise *entails*, *contradicts*, or is *neutral* with respect to a hypothesis. Yet, ambiguities in the premise and/or hypothesis (as in Table 1) may impact the determination of the label.

We present AMBIENT, a dataset of 1,645 NLI examples, each annotated with a *set* of labels, reflecting potentially multiple readings of the premise and/or hypothesis. Ambiguous examples, i.e., those having more than one label, make up 35.2% of the dataset and include a **disambiguating rewrite** corresponding to each label; unambiguous examples have a single label. The inclusion of unambiguous examples facilitates evaluating model abilities to first detect the presence of relevant ambiguity, and then resolve it to distinct interpretations.

We use two approaches to collect source examples: manual curation and automatic generation. Manual curation (§2.1) involves crafting a small set of examples targeting specific types of ambiguity. Further, to cover more diverse forms of ambiguity, we produce a larger collection of examples via text generation and heuristic filtering (§2.2), followed by expert manual annotation (§2.3), forming the bulk of AMBIENT. Details are in §A.

### 2.1  Curated Examples

The authors curate a set of 142 examples, which are either handwritten or sourced from existing NLI datasets and linguistics textbooks (Kearns, 2000;

---

[2]For example, Koller et al. (2008) find that including all possible quantifier scope readings in the Rondane Treebank (Copestake and Flickinger, 2000) results in 5% of sentences having ≥650,000 possible semantic analyses.

[3]By design, we only consider ambiguities in NLI examples that affect the determination of the label, to provide a natural setting where ambiguities are contextually relevant.

| Example | Disambiguation 1 | Disambiguation 2 | Type |
|---|---|---|---|
| P: I'm afraid the cat was hit by a car.
H: The cat was not hit by a car.
⟨NEUTRAL, CONTRADICT⟩ 👤: [**7** N, **2** C] | P: I'm worried...
NEUTRAL 👤: [**9** N] | P: I'm sorry to share that...
CONTRADICT 👤: [**9** C] | *Pragmatic* (44.8%) |
| P: John and Anna are married.
H: John and Anna are not a couple.
⟨NEUTRAL, CONTRADICT⟩ 👤: [**5** N, **4** C] | P: ... are both married.
NEUTRAL 👤: [**7** N, **2** E] | P: ... are married to each other.
CONTRADICT 👤: [**9** C] | *Lexical* (20.0%) |
| P: This seminar is full now, but interesting seminars are being offered next quarter too.
H: There will be more interesting seminars...
⟨ENTAIL, NEUTRAL⟩ 👤: [**7** E, **2** N] | H: There will be more seminars ... that are interesting.
ENTAIL 👤: [**9** E] | H: There will be seminars... that are more interesting.
NEUTRAL 👤: [**9** N] | *Syntactic* (8.6%) |
| P: The novel has been banned in many schools because of its explicit language.
H: The novel has not been banned in many schools.
⟨NEUTRAL, CONTRADICT⟩ 👤: [**4** N, **5** C] | H: There are many schools where the novel has not been banned.
NEUTRAL 👤: [**9** N] | H: It is not the case that the novel has been banned in many schools.
CONTRADICT 👤: [**9** C] | *Scopal* (7.6%) |
| P: It is currently March, and they plan to schedule their wedding for next December.
H: They plan to schedule... for next year.
⟨ENTAIL, CONTRADICT⟩ 👤: [**3** E, **2** N, **4** C] | P: ... for December next year.
ENTAIL 👤: [**9** E] | P: ... for the coming December.
CONTRADICT 👤: [**9** C] | *Coreference* (2.9%) |
| P: It is difficult to believe that the author of such a masterpiece could have been only 23 years old.
H: The author of the masterpiece was only 23.
⟨ENTAIL, NEUTRAL⟩ 👤: [**3** E, **6** N] | P: It is shocking that...
ENTAIL 👤: [**9** E] | P: It is questionable that...
NEUTRAL 👤: [**9** N] | *Figurative* (1.9%) |
| P: A new study has found that nearly half of all Americans are in favor of gun control.
H: The study found that half of all Americans are in favor of gun control.
⟨ENTAIL, CONTRADICT⟩ 👤: [**1** E, **2** N, **6** C] | H: ... that exactly half of all Americans...
CONTRADICT 👤: [**8** C, **1** N] | H: ... that about half of all Americans...
ENTAIL 👤: [**9** E] | *Other* (14.3%) |

Table 1: Ambiguous examples in AMBIENT with linguist-annotated ⟨**GOLD LABELS**⟩. As analysis, we collect the 👤: [distribution of NLI labels] as judged by nine crowdworkers under the traditional single-label annotation scheme (§3), finding that disagreement on ambiguous examples is largely resolved on disambiguations. The **Type** column indicates the ambiguity type for each example, along with its estimated representation in the dataset (§2.5).

Carnie, 2013). We choose examples ad hoc from the synthetic NLI datasets **DistNLI** (Ban et al., 2022) for predicate distributivity (e.g., "*Sam and Frank gave a talk*" may either mean separately or jointly) and **ImpPres** (Jeretic et al., 2020) for implicatures. We also include some instances with differing pragmatic and literal readings from **NLI Diagnostics** (Wang et al., 2018), and ones leading to disagreement from large-scale NLI datasets like **MNLI** (Williams et al., 2018) and **WANLI** (Liu et al., 2022). The authors directly annotate these examples with the set of labels and disambiguations (examples in §A.1).

## 2.2 Generated Examples

To cover more ambiguities, we use overgeneration and filtering to automatically create a large corpus of unlabeled NLI examples that are likely to be ambiguous. Inspired by WANLI (Liu et al., 2022), we automatically identify groups of premise-hypothesis pairs that share a reasoning pattern, to encourage the creation of new examples with the same pattern. We use WANLI as our source of examples; each group contains a randomly chosen example on which its two annotators disagreed (indicating possible ambiguity), along with its 4 nearest neighbors according to the final-layer embedding of a WANLI-trained NLI model. We observe that these groups can share interpretable ambiguity patterns, such as sentences about the past (e.g., "*When I was young, I was obsessed*") inducing a cancellable implicature about the present (that "*I*" am no longer obsessed; full prompt in §A).

These groups of examples are formatted into a prompt with the instruction, "*Write pairs of sentences that are related to each other in the same*

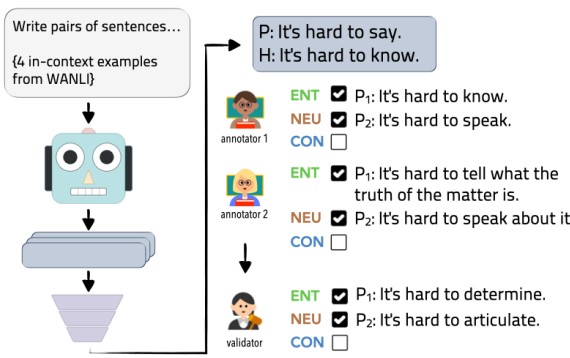

Figure 2: Pipeline for the annotation of generated examples in AMBIENT. Unlabeled examples are created by `InstructGPT`, then annotated independently by two linguists, whose annotations are consolidated by an author.

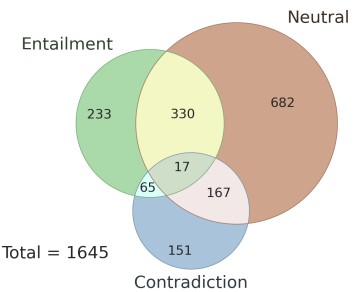

Figure 3: Distribution of set labels in AMBIENT.

*way.*" For each prompt, we sample 5 continuations from `InstructGPT` (Ouyang et al., 2022), discarding those that cannot be parsed into a premise and hypothesis.

To further filter for likely-ambiguous instances, we use a multilabel RoBERTa-large model trained on WANLI and retain all examples where the model assigns probability ≥ 0.05 to more than one NLI label, indicating at least slight uncertainty in whether there can be multiple possible readings.

## 2.3 Annotation and Validation

Examples acquired in §2.2 consist of unlabeled premise-hypothesis pairs, which we next annotate with label sets and relevant disambiguations. Following AMBIGQA (Min et al., 2020) and as shown in Figure 2, each example is first annotated by two experts, then presented to a third expert for validation and consolidation.

We recruit 37 university-level linguistics students for the annotation phase,[4] as identifying ambiguities of a sentence then delineating its possible interpretations is a challenging task. They select a set of labels for each example, including the singleton set when the example is unambiguous; when more than one label is chosen, they provide a disambiguating rewrite for each one. They are asked to discard the example if it is offensive or low-quality due to issues in fluency or coherence.

The validation phase is performed by a subset of the authors to ensure high quality (details in §A.4). The authors review the two sets of annotations to revise and aggregate them into a single coherent annotation, optionally adding interpretations missed by both annotators. Validation is skipped when ei-

---

[4]We refer to them as "linguists" elsewhere.

ther annotator discarded an example; the validators may additionally discard examples themselves.

Linguists annotate a total of 2,616 examples. Due to the option for discarding, 2,020 examples emerge from the annotation phase, and after validation, there are a total of 1,503 final examples.

## 2.4 Agreement

To calculate inter-annotator agreement for validation, the four validators annotate a subset of 50 examples in common. The Fleiss $\kappa$ agreement score on the binary classification task for each label is 0.62 for contradiction, 0.65 for entailment, and 0.44 for neutral, thus ranging from "moderate" to "substantial" agreement.

## 2.5 AMBIENT Statistics

The final dataset, which combines curated and generated-then-annotated examples, consists of 1,645 examples. We sample 100 for a development set and treat the rest as the test set. The label distribution is shown in Figure 3.

To understand the types of ambiguity present in AMBIENT, the authors annotate a random subset of 100 ambiguous examples with the ambiguity type, among *lexical*, *syntactic*, *figurative*, *pragmatic*, *scopal*, *coreference*, and *other* (described in §A.6). Results are shown in the **Type** column of Table 1.

## 3 Does Ambiguity Explain Disagreement?

We conduct an analysis to understand how annotators behave on ambiguous input, under the traditional 3-way annotation scheme for NLI. We find that ambiguity is recognizable to individual workers and explains much of the label variation that emerges, thus challenging the popular assumption that example uncertainty should be modeled as "disagreement" among annotators.

### 3.1 Setup

We recruit crowdworkers on Amazon Mechanical Turk to review ambiguous examples in AMBIENT.

Each example is reviewed by 9 workers. The task is split into three steps, each appearing only after the earlier steps are complete.

**(i) Annotation of ambiguous example** Following the traditional NLI labeling setup, crowdworkers are presented with the original ambiguous example alone, and asked to choose a single label.

**(ii) Recognition of disambiguations** The ambiguous sentence of the example (either the premise or hypothesis) is isolated for consideration.[5] Three candidate interpretations are presented in a random order, composed of the two disambiguations and a semantically similar "distractor". (In the case where an example has three interpretations, no distractor is included.) Workers are asked to indicate whether each sentence is a "possible interpretation" of the isolated sentence. We instruct that this is subjective, and they should use their best judgment.

The distractor ensures that workers do not consider all sentences as valid readings, and is obtained by back-translating the ambiguous sentence with Yorùbá using NLLB (Meta, 2022). A low-resource language is chosen so that the back-translation is a close, but often not entirely faithful, paraphrase.

**(iii) Annotation of disambiguated examples** Three new NLI examples are obtained by substituting the ambiguous sentence of the original example with each candidate interpretation from **(ii)**. Workers select a single NLI label for each new example.

### 3.2 Results

As hypothesized, the original ambiguous examples produce high disagreement, with a Fleiss $\kappa$ score of 0.12, considered "slight" agreement (step (i)). Disagreement is largely resolved on the corresponding disambiguated examples (step (iii)), with $\kappa$ increasing to 0.67, representing "substantial" agreement.

Moreover, annotators overwhelmingly recognize disambiguations as plausible interpretations of the ambiguous sentence (step (ii)). True disambiguations are marked plausible 96.7% of the time, compared to 46.7% for the distractor. On average, 93.7% of annotators accept *all* true interpretations, thus recognizing the full set of possibilities.

Through this experiment, we additionally establish crowdworker agreement with AMBIENT as the rate at which the *majority vote* recognizes the full

set of ambiguities (step (ii)) and verifies their labels (step (iii)). In this sense, the agreement rate is 89.7%. This points to the quality of the dataset and is used as a reference point for later experiments.

Overall, input ambiguity is indeed a source of "disagreement" in NLI under a single-label annotation scheme. However, we have shown that *individual* annotators overwhelmingly can recognize *multiple* possible readings of the input and their corresponding output labels, and much of this disagreement can be resolved in practice by incorporating disambiguation into the task. In this way, input ambiguity can be disentangled from annotator subjectivity.

## 4 Evaluating Pretrained Language Models

In our experiments, we investigate the extent to which understanding of ambiguity is acquired during the course of pretraining. Our three tests evaluate if LMs can directly **generate** relevant disambiguations (§4.1), **recognize** the validity of plausible interpretations (§4.2), and finally, **model** open-ended continuations reflecting different interpretations (§4.3). For these tests, we consider only the ambiguous instances in AMBIENT.

As our set of LMs, we evaluate LLaMa (65B; Touvron et al., 2023) and GPT-3 (davinci), as well as instruction-tuned models FLAN-T5 (xxl; Chung et al., 2022), InstructGPT (text-davinci-003), ChatGPT (gpt-3.5-turbo), and the recent GPT-4.

### 4.1 Generating Disambiguations

We first study whether LMs can learn in-context to directly generate disambiguations and corresponding labels. We construct a natural prompt (see Table 2) by explaining that there is some ambiguity that makes the correctness of a "*claim*" (hypothesis) difficult to resolve given the "*context*" (premise). For each test instance, we randomly sample 4 other test instances as in-context examples.

As there are multiple ways to express the same disambiguation, we perform both automatic and human evaluation. For the former, we match each generated disambiguation with a reference disambiguation based on the generated label.[6] Following AMBIGQA, we score generations using the EDIT-F1 metric, which represents a disambiguation by

---

[5]For simplicity, we only include examples where *either* the premise *or* the hypothesis is ambiguous (93.1% of examples).

[6]If the label verbalizer for a disambiguation does not correspond to any label in the reference label set, then the model receives a score of 0 for that disambiguation.

its added and deleted unigrams, and computes the F1 score between the reference and the prediction.

For human evaluation, we use the same setup as the crowdworker experiment in §3 on 50 randomly sampled examples, except without step (i). We use three workers per example, and consider the LM correct on an example if the majority vote indicates that each disambiguation is plausible (step (ii)) *and* selects the model-predicted NLI labels (step (iii)). Crowdworkers are not informed that disambiguations are model-generated.

**Results**   Shown in Table 4, the best model is GPT-4, achieving an EDIT-F1 score of 18.0% and human-judged correctness of 32.0%. The latter can be directly compared to crowdworker agreement with AMBIENT itself at 89.7% (§3).

One strategy for attempting disambiguation we observe across models is restating the ambiguous sentence with additional context that directly affirms or negates the hypothesis, rather than making a targeted revision to clarify the ambiguity. In some cases, this "shortcut" does lead to technically correct disambiguations (and marked as such in human evaluation). For instance, for

> P: He always ignores his mother's advice to follow his own dreams.
> H: He follows his dreams.

ChatGPT disambiguates the premise by restating it, followed by "*and therefore does follow his dreams*" versus "*and therefore does not follow his dreams*." The former forces the interpretation that he ignores her advice *in order to* follow his dreams; the latter the interpretation that his mother's advice *is* for him to follow his dreams. Thus, the human-judged correctness may overestimate the models' ability to precisely report the source of ambiguity.

## 4.2   Recognizing Disambiguations

For the next test, we focus on the ambiguous sentences alone (without the rest of the NLI example), and create a series of templated true and false statements about possible interpretations as shown in Table 3. For instance, it is both true that an ambiguous sentence *may mean* a particular interpretation, but also that it does not *necessarily* mean it. We consider the model prediction to be the token with the greater logit between True and False.[7] We

---

[7]As the API for ChatGPT and GPT-4 does not return token logits, we simply consider whether the top-1 token is correct. We find either True or False is the top token in 97.6% and 99.7% of examples, respectively, indicating the task is clear.

**Instruction**
In each example, you will be given some **context** and a **claim**, where the correctness of the **claim** is affected by some ambiguity in the **context**. Enumerate two or three interpretations of the **context** that lead to different judgments about the **claim**.

**Example**
**Context**: {premise}
**Claim**: {hypothesis} Given the context alone, is this **claim** **true**, **false**, or **inconclusive**?
We don't know, because the **context** can be interpreted in many different ways:
1. {disambiguation 1} Then the **claim** is **true**.
2. {disambiguation 2} Then the **claim** is **false**.
3. {disambiguation 3} Then the **claim** is **inconclusive**.

Table 2: Few-shot template for the task of generating disambiguations (§4.1) when the *premise* is ambiguous. The label verbalizer correspondences are true ↔ ENTAIL, false ↔ CONTRADICT, and inconclusive ↔ NEUTRAL. The instruction is stated once, followed by four in-context examples. At the end of the prompt, the test example is provided up until "*1.*".

| Template | Correct Answer |
|---|---|
| {a} This may mean: {d} | True |
| {a} This does not necessarily mean: {d} | True |
| {a} This cannot mean: {d} | False |
| {a} This can only mean: {d} | False |

Table 3: Templates for True/False evaluation (§4.2), where {a} denotes the ambiguous sentence and {d} a possible disambiguation. Given the infilled template followed by "*True or False?↵Answer:*", the LM is expected to choose the correct answer.

execute this task zero-shot as the prompt template completely determines the label.

**Results**   The **T/F Acc.** column of Table 4 shows the accuracy averaged across the four templates. The best model (GPT-4) achieves only 63.0% compared to the random accuracy of 50%, with other models ranging between 49.6% and 57.7%. When we consider the proportion of disambiguations for which GPT-4 answers all four templates correctly, performance drops to 2.5%, below random guessing of 6.25%. We do not observe consistent trends across models on the per-template accuracy (shown in §C.2), though four of six models achieve the highest accuracy on template 1.

Furthermore, we observe that LMs are not *internally* consistent across the questions. For example, for 76% of pairs of disambiguations $(d_1, d_2)$ for the same ambiguous sentence $a$, GPT-4 both acknowledges that $a$ may mean $d_1$ and may mean $d_2$ (template 1), yet also asserts that $a$ can *only* mean $d_1$ and can only mean $d_2$ (template 4).

|            | Edit-F1 | Correct (human) | T/F Acc. | KL Rank. Acc. |
|------------|---------|-----------------|----------|---------------|
| FLAN-T5    | 5.2     | 0.0             | 56.4     | **81.0**      |
| LLaMa      | 10.0    | 10.0            | 55.0     | 68.9          |
| GPT-3      | 10.1    | 2.0             | 57.8     | 75.7          |
| InstructGPT| 14.5    | 4.0             | 49.6     | 71.4          |
| ChatGPT    | 13.0    | 18.0            | 57.7     | -             |
| GPT-4      | **18.0**| **32.0**        | **63.0** | -             |

Table 4: Performance of pretrained models on AMBIENT. Higher values are better for all metrics. A baseline that reproduces the ambiguous sentence as its disambiguation would achieve 0 Edit-F1 and human-judged correctness; random performance for T/F accuracy is 50% and for KL ranking accuracy is 32.8%.

### 4.3 Modeling Interpretation-Specific Continuations

Finally, we determine whether LMs, when conditioned on an ambiguous sentence, implicitly model different interpretations in their distributions of text continuations. Since LMs are trained to model words given context, understanding ambiguity should mean recognizing the *union* of the contexts for a sentence's interpretations.

To measure this, we obtain continuations for each interpretation, and quantify how "surprised" the LM is to see them when conditioned on the ambiguous sentence.[8] Specifically, we first sample 100 continuations $c \sim P(\cdot \mid d_i)$ conditioned on each *disambiguation* $d_i$ as context. Then, we compare the likelihood of $c$ under the ambiguous sentence $a$ versus the corresponding disambiguation $d_i$ by computing $\log P(c \mid d_i) - \log P(c \mid a)$. This describes how much the LM "suffers" by seeing the ambiguous instead of the unambiguous context,[9] and is an unbiased estimate of the KL divergence between $P(\cdot \mid d_i)$ and $P(\cdot \mid a)$ (proof in §C.3):

$$D(P(\cdot \mid d_i) \mid\mid P(\cdot \mid a))$$
$$= \lim_{N \to \infty} \frac{1}{N} \sum_{\substack{j=1 \\ c_j \sim P(\cdot \mid d_i)}}^{N} \log \frac{P(c_j \mid d_i)}{P(c_j \mid a)}.$$

Intuitively, we want the KL divergence not to be too large — the LM should reasonably expect to see continuations for either interpretation. To quantify this, we introduce a "distractor" sentence $\tilde{d}$ formed by replacing a randomly selected noun in $a$ with a same-category word from ConceptNet (Speer et al.,

2017), e.g., replacing "*school*" with "*library*."

We expect the LM to model continuations from *both* disambiguations $d_i$ better than those from the distractor $\tilde{d}$, i.e., for all true disambiguations $d_i$,

$$D(P(\cdot \mid \tilde{d}) \mid\mid P(\cdot \mid a)) > D(P(\cdot \mid d_i) \mid\mid P(\cdot \mid a)).$$

We call the fraction of ambiguous contexts for which this is true the **KL ranking accuracy**.

**Results** The **KL Rank. Acc.** column of Table 4 shows that FLAN-T5 demonstrates the correct preference of continuations for 81.0% of examples, making it the best model here despite its poor performance in other settings. The inconsistent trends suggest that results are heavily dependent on how competence on ambiguity is operationalized. Nonetheless, ambiguity remains a severe challenge across models and across the suite of tests.

## 5 Evaluating Multilabel NLI Models

Given that language models still struggle to process ambiguity in §4, we next investigate the effectiveness of finetuning them on existing NLI data collected in the line of work on underspecification and subjectivity in NLI.[10] Here, we consider the discriminative task of multilabel NLI prediction, across both ambiguous and unambiguous examples in AMBIENT. Experimental details are in §D.

### 5.1 Methods

We experiment with methods that predict a single probability value, a distribution over labels, or a set of labels. We use the development set of AMBIENT to tune threshold(s) that map the output of these models onto a set of labels (see §D.1). All models are based on roberta-large, and we report results over 5 random seeds for model training.

**Regression models** We train a regression model on **Uncertain NLI** (UNLI; Chen et al., 2020) that predicts a value on $[0, 1]$ representing the probability of the hypothesis being true given the premise.

**Distributional models** Distributional models aim to predict the distribution of annotator judgments. We use two models from prior work: 1) one trained on **AmbiNLI** (Meissner et al., 2021), with examples with multiple annotations from SNLI (Bowman et al., 2015) and MNLI, and 2)

---

[8]We exclude ChatGPT and GPT-4 from evaluation as the API does not enable calculating likelihood under the model.

[9]This method assumes that the likelihood of a continuation is based on its meaning alone, but surface-form attributes like style are a confounding factor. See further discussion in §C.3.

[10]The size of AMBIENT is not large enough for a training split; future efforts to annotate data in the fashion of AMBIENT might be able to address this issue.

| Method and Train Set | EM | Macro F1 | Group EM |
|---|---|---|---|
| **Reg.** Uncertain NLI (C+20) | $24.5_{2.3}$ | $62.2_{1.0}$ | $4.7_{2.5}$ |
| **Dist.** AmbiNLI (M+21) | $21.0_{1.6}$ | $63.8_{0.8}$ | $10.1_{2.5}$ |
| SNLI + MNLI (Z+22) | $24.3_{1.1}$ | $68.0_{0.1}$ | $4.7_{1.2}$ |
| **Multi.** Multi-label MNLI (JD22) | $15.8_{3.4}$ | $63.2_{0.6}$ | $0.9_{1.2}$ |
| Multi-label WANLI (L+22) | $35.1_{3.0}$ | $\mathbf{72.5}_{0.3}$ | $19.1_{4.8}$ |
| Classifier over sets WANLI | $\mathbf{43.6}_{0.8}$ | $70.7_{0.2}$ | $\mathbf{37.8}_{0.4}$ |

Table 5: Performance of multilabel NLI models on AMBIENT. While all model outputs are mapped onto a label set, their original output is one of regression (reg.), distributional (dist.), or multilabel (multi.) output. **EM** and **Macro F1** measure performance on the original example; **group EM** considers performance on both the original example and its disambiguations. We report the mean and standard deviation over 5 seeds.

a model trained through **distribution distillation** (Zhou et al., 2022), where a teacher model trained on SNLI + MNLI is used to re-annotate the data with soft labels then used to train a new model.

**Multilabel models** Prior work trained a **multilabel** model (Jiang and de Marneffe, 2022) on the development sets of MNLI + ChaosNLI by turning distributional labels into discrete ones with a threshold of 0.2. In addition, we train a multilabel model on WANLI's train set (which has two annotations per example), as well as a **classifier over sets** which performs 7-way classification over the power set of NLI labels, minus the empty set.

### 5.2 Metrics

On the original examples, we calculate the **macro F1** score and the **exact match** accuracy (EM); the latter requires the model to exactly predict the label set. We also report the **group EM** accuracy as the fraction of examples where the model exactly predicts the label set for both the original NLI example *and* all of its disambiguations.

### 5.3 Results

As shown in Table 5, the multilabel model trained on WANLI achieves the highest macro F1 score of 72.5%, and the classifier over sets achieves the best EM accuracy of 43.6% and group EM accuracy of 37.8%. While the EM accuracy is substantially higher than the random-guessing baseline of 1/7 = 14.3%, it is considerably short of 89.7%, the rate at which crowdworkers correctly predict the set of labels when presented with possible disambiguations (§3). Overall, finetuning NLI models on existing data with label variation still leaves large room for improvement on the multilabel NLI task.

## 6 Case Study: Detecting Misleading Political Claims

We illustrate the value of ambiguity-sensitive models via a case study in detecting misleading political claims in the wild. Here, we use the key insight that *for ambiguous sentences, some paraphrases are naturally disambiguating*, as paraphrases must either preserve the ambiguity or paraphrase a particular interpretation. Therefore, if we cast a given sentence as the premise and a paraphrase as the hypothesis, a multilabel NLI model predicting two or more labels should indicate the presence of ambiguity. Moreover, the paraphrase resulting in this prediction should reveal the source of ambiguity.

We experimentally evaluate this idea on the development set of CLAIMDECOMP (Chen et al., 2022), which contains 200 claims with their PolitiFact fact-checks. The authors read each instance and mark whether the fact-check describes an issue of ambiguity or factuality (regardless of whether we perceive ambiguity ourselves). Then we paraphrase each claim 5 times with InstructGPT zero-shot, and apply the **multilabel WANLI** model from §5, which achieved the highest F1 score, on each resulting NLI example. A claim is considered ambiguous if the model predicts more than one label for any paraphrase. Examples in Table 6.

This method recalls 88.8% of ambiguous claims. While precision is lower at 12.4%, qualitative inspection of false positives reveals many ambiguities that were left unmentioned in the fact-check, illustrating the potential of these tools to anticipate sources of misunderstanding. Ultimately, our analysis suggests that fact-checking as a more general problem may need refinement, due to the possible presence of both true and false interpretations. This case study shows only one use case of ambiguity-sensitive models, and we hope for AMBIENT for benchmark further progress on this front.

## 7 Related Work

**Ambiguity** Ambiguity is a longstanding and well-studied issue for NLP tasks involving symbolic analyses of sentences, such as syntactic and semantic parsing (Church and Patil, 1982; Koller et al., 2008) or coreference resolution (Poesio and Artstein, 2005). However, as the field has recently shifted toward higher-level understanding and reasoning problems, ambiguity in language has been largely overlooked.

In the space of open-domain question-answering,

| Political claim (premise) | Generated paraphrase (hypothesis) | Rating | Prediction | Explanation of ambiguity (ours) |
|---|---|---|---|---|
| When President Obama was elected, the market crashed... | The stock market reacted immediately to President Obama's election in 2008, ... | Barely-true | ⟨ENTAIL, NEUTRAL⟩ | The claim implies a causal relationship |
| Rhode Island is "almost dead last"... in the length of time first-degree murderers must spend in prison before they're eligible for parole. | Rhode Island is one of the states... where murderers must spend the longest time in prison before being eligible for parole. | True | ⟨ENTAIL, NEUTRAL, CONTRADICT⟩ | "*dead last*" may mean shortest or longest, depending on stance |
| Donald Trump even said, on his very first day in office, he would require every school in America to let people carry guns into our classrooms. | Donald Trump said on his first day in office that every school in America would have to allow people to carry guns in classrooms. | True | ⟨ENTAIL, NEUTRAL⟩ | "*on his first day*" may describe either the *saying* or the *requiring* |

Table 6: Political claims flagged as ambiguous by our detection method. For the claim in the first row, the ambiguity was noted by the fact checker (**Rating** column), thus leading to a barely-true rating; in the bottom two, the ambiguity was not mentioned, showing the value of this method for ambiguity detection.

there are often issues of ambiguous or under-specified event and entity references (Min et al., 2020; Cole et al., 2023), leading to work on generating clarification questions (Kuhn et al., 2023; Krasheninnikov et al., 2022). In particular, our approach to ambiguity is inspired by AMBIGQA (Min et al., 2020), where the task input is disambiguated in natural language to account for variation in possible outputs. In contrast to open-domain questions, AMBIENT contains more diverse linguistic ambiguities whose resolution is a prerequisite to understanding meaning.

Recent work has also studied ambiguous language in multi-modal settings: Stengel-Eskin et al. (2023) collected a set of ambiguous questions about images, and Pezzelle (2023) consider how vision-language models handle underspecified captions.

Other work studies whether the confidence of coreference and NLI models is sensitive to ambiguity in synthetically-constructed input (Yuan et al., 2023; Ban et al., 2022). Going beyond task-specific models, we evaluate *pretrained* LMs for the language skill of managing ambiguity.

**Human label variation** Human label variation (Plank, 2022) is a broad phenomenon with three distinct sources, as summarized by Jiang and de Marneffe (2022): *task* ambiguity, *subjectivity* of annotator attitudes, and *input* ambiguity (our focus). Explored in contemporary work (Tamkin et al., 2023), task ambiguity arises when the task is underspecified with respect to the desired output; subjectivity is observed when different people disagree, such as for toxic language detection (Sap et al., 2022).

There is growing recognition of and interest in studying this variation, where the dominant approach is to model the *distribution* of human judgments (Pavlick and Kwiatkowski, 2019; Nie et al.,

2020; Uma et al., 2021), potentially as a function of their demographic characteristics (Gordon et al., 2022). In our work, we argue that when uncertainty is in the input, we should instead directly characterize the underlying reasons for the uncertainty.

**NLI beyond three-way classification** For NLI, the seminal work investigating label variation was Pavlick and Kwiatkowski (2019), and subsequent work collected more annotations (Nie et al., 2020) and modeled this variation (Zhou et al., 2022; Zhang et al., 2021). Other approaches aim to predict the probability of entailment (Chen et al., 2020; Zhang et al., 2017) or a fourth "disagreement" label (Zhang and de Marneffe, 2021). We contribute another approach, where NLI models predict the *set* of labels for plausible readings.

Jiang and de Marneffe (2022) investigate MNLI data to taxonomize sources of disagreement, and identify "uncertainty in sentence meaning" as one source, though they named only lexical and implicature ambiguities. Our benchmark includes a wider coverage of ambiguities and our analysis further sheds light on the nature of the "disagreement."

## 8 Conclusion

Ambiguity in language will become increasingly conspicuous as we push the limits of LM capabilities and build tools that engage with the nuances of natural language communication. We develop the first benchmark to evaluate whether language models recognize different readings of ambiguous text, and demonstrate that the task remains extremely challenging. We encourage future work to study the sensitivity of LMs to context and emphasis, investigate the presence of systematic biases in interpretation, and explore the promising space of real-world applications enabled by ambiguity-sensitive tools.

## Acknowledgments

We would like to thank Nathan Schneider, Ellie Pavlick, Doug Downey, Ewin Tang, Roy Schwartz, Yanai Elazar, Valentina Pyatkin, and Ari Holtzman, as well as the greater UW NLP and AI2 community, for valuable feedback and discussion at different stages of this work. Our dataset would not have been possible without the expertise of our linguist annotators, which include Emma Miller, Sofia Y. Ahmed, Wendy Kempsell Jacinto, Maxine Appel, Edi Xin, Magdelina Thornton, Huijae Seo, Gita Dhungana, and Aldrich Gran Lapid, and 28 others.

This work was funded in part by the DARPA MCS program through NIWC Pacific (N66001-19-2-4031). We thank OpenAI for offering access to various models through the API. The first author is supported by the National Science Foundation Graduate Research Fellowship Program.

## Limitations

In this work we collect a broad-coverage dataset of ambiguities, but the size and diversity are nonetheless limited due to the data sources and the effort required for expert annotation. We thus encourage future work to collect more data in the format of AMBIENT, especially for naturally-occurring ambiguities. In addition, we only study ambiguity phenomena in English, but how ambiguity manifests in other languages can vary greatly due to systematic typological factors or idiosyncratic differences. For example, while AMBIENT does not contain many instances of morphological ambiguity, these are very common in morphologically richer languages such as Turkish and Finnish. A systematic extension of our dataset and analyses to other languages would be exciting future work.

Though LMs struggle across the board on our evaluations, this does not guarantee that they will not handle ambiguity well in other task settings or using other extraction methods. We observe that GPT-4 is the highest-performing model on two of the three evaluations (§4.1, §4.2), while the smallest FLAN-T5 performs best on the last evaluation (§4.3). Scaling up general-purpose pretraining and reinforcement learning from human feedback (Ouyang et al., 2022) may lead to further gains, though we hypothesize that the trend will be unclear as larger LMs may overfit to more common interpretations at the expense of recognizing less common ones, which is especially detrimental for reasoning about misleading language.

## Ethics Statement

We acknowledge that text generated from language models is susceptible to perpetuating social harms and containing toxic language (Sheng et al., 2019; Gehman et al., 2020). To address this, the annotators and validators of our dataset (§2.3) were instructed to discard any examples that may be perceived as offensive. Nonetheless, it is possible that subtly harmful examples may have been overlooked and included in the final dataset.

In addition, we are cognizant of the asymmetrical relationship between requesters and workers in crowdsourcing (§3). We took great care to pay fair wages, with a median hourly rate of $19.13, and were responsive to feedback and questions throughout the process (see §B.1 for details). The only personal information we collect is the worker IDs from Amazon Mechanical Turk, which we will not release. Both the linguist annotation (§2.3) and crowdworker experiment (§3) received IRB exemption.

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

## A  Dataset Creation Details

### A.1  Curated Examples

The first author skimmed through several existing NLI datasets and manually identified examples that were both natural and contained salient ambiguities. Only a few examples were chosen from each dataset, to avoid overly redundant examples in AMBIENT. In Table 7, we show an example from each of the sources we drew from. They are directly annotated with the set of labels and disambiguations by the first author.

### A.2  Generated Examples

The template for prompting GPT-3 to generate unlabeled NLI examples is shown in Table 8. The model we used is InstructGPT (text-davinci-002), queried on September 4, 2022, with top $p = 0.9$ (Holtzman et al., 2020), max tokens 120, and stop sequence "\n\n". If the generated output is not correctly formatted with

"\nSentence 2:" in the sequence (which separates the premise and hypothesis), we discard the output. We sample 5 outputs per 21,273 possible prompts to obtain a total of 104,071 unlabeled examples.

We first employ simple heuristics to discard examples exhibiting observable failure cases. That is, we discard examples if 1) either the premise or hypothesis is shorter than 5 characters, 2) the premise and hypothesis are identical, 3) the generated example is copied from an in-context example, or 4) the examples contain some redundant patterns observed in the development phase. For instance, there are an abundance of generations with the exact premise, "*Mary wants to try a little bit of every country's food on her trip around the world.*" After filtering based on these rules, 77,564 examples remain.

Next, to further filter for likely-ambiguous instances, we use a multilabel RoBERTa-large model trained on WANLI and retain all examples where the model assigns probability $\geq 0.05$ to more than one NLI label, indicating at least slight uncertainty in whether there can be multiple possible readings. Finally, to approximately balance the resulting examples (of course, exactly balancing would be impossible without gold labels), we keep an equal number of examples where the multilabel model predicts (according to the low threshold of $0.05$) ⦃ENTAIL, NEUTRAL⦄ and ⦃CONTRADICT, NEUTRAL⦄, and all other examples with multiple labels predicted. Thus, the final set of generated examples is 16,826.

### A.3  Linguist annotation

Of the generated examples, we ultimately annotate only 2,616 of them. This is due to the slow pace of expert annotation as the project went on, and the diminishing returns of annotating more data. Each example was annotated by two linguists. We discard an example if either linguist chose to discard it. The final set of examples is 2,020.

Our expert annotators were 37 university-level linguistics students at the University of Washington, recruited through a Linguistics mailing list. They were paid $20/hour, in addition to $0.05 per example.

### A.4  Validation by authors

The authors review all 2,020 examples, each with two annotations, combining and revising them into a single coherent annotation, or optionally discarding the example. The authors validated the ex-

| Example | Disambiguation 1 | Disambiguation 2 | Source |
|---|---|---|---|
| P: It is the only possibility of making the law the servant of the people, not the other way around.
H: The law should be the servant of the people, not the other way around.
⦃ENTAIL, NEUTRAL⦄ | P: ... of making the law the servant of the people, as it should be, ...
ENTAIL | P: It is the only possibility that would lead to the law...
NEUTRAL | WANLI test |
| P: Then he sobered.
H: He was drunk.
⦃ENTAIL, NEUTRAL⦄ | P: Then he sobered after drinking alcohol.
ENTAIL | P: Then he became more sensible.
NEUTRAL | MNLI dev |
| P: Patrick did not manage to leave.
H: Patrick tried to leave.
⦃ENTAIL, NEUTRAL⦄ | P: ..., despite his attempt.
ENTAIL | P: ... , whether or not he tried.
NEUTRAL | IMPPRES |
| P: LaBeouf had tried to bum a smoke from two strangers, unaware that one of them was a police officer.
H: LaBeouf had tried to bum a smoke from a police officer.
⦃ENTAIL, CONTRADICTION⦄ | H: LaBeouf had tried to find a police officer to bum a smoke from.
ENTAIL | H: LeBeouf had tried to bum a smoke from someone who happened to be a police officer.
CONTRADICTION | NLI Diagnostics |
| P: Jenny and Zoe solved the puzzle.
H: They solved it together.
⦃ENTAIL, CONTRADICTION⦄ | P: ... solved the puzzle together.
ENTAIL | P: ... each solved the puzzle.
CONTRADICTION | DistNLI |
| P: John opened the door again.
H: John opened the door before.
⦃ENTAIL, NEUTRAL⦄ | P: John opened the door before, and did it again.
ENTAIL | P: The door was open before, and John opened the door again.
NEUTRAL | Carnie |
| P: John wishes to marry Adrienne, a Frenchwoman.
H: John wants to marry a Frenchwoman.
⦃ENTAIL, NEUTRAL⦄ | P: John wants to marry a certain woman who is French.
ENTAIL | P: John wants for his future wife to be French.
NEUTRAL | Kearns |
| P: You should visit Norway in the summer.
H: Summer is a good season to visit Norway.
⦃ENTAIL, NEUTRAL⦄ | P: You should visit Norway the coming summer.
ENTAIL | P: You should visit Norway in the summer season.
NEUTRAL | Handwritten |

Table 7: An example in AMBIENT from each of the sources we draw from for the curated examples (§2.1).

amples together on Zoom calls over the course of several weeks, actively discussing examples that they were unsure about and developing consistent standards. For instance, we chose to discard examples that boiled down to temporal ordering (e.g., "*I didn't realize that I left my keys at home*" either entails or contradicts "*I realized I left my keys at home*", depending on the ordering of the sentences) or vagueness (e.g., "*He is six feet tall*" may or may not entail "*He is tall*" due to the vagueness of the word "*tall*"). This process revealed to us the extent of task underspecification in NLI (something we do not directly study in this work), and allowed us to focus on linguistic ambiguity.

Ultimately, 1,503 examples emerge from this phase.

## A.5 Additional statistics

The disambiguating rewrites are, on average, 2.36 words longer than their ambiguous counterparts. Among the ambiguous examples, 74.3% have ambiguity in the premise and 32.6% in the hypothesis, with 6.9% having ambiguity in both. 97.5% of ambiguous sentences are labeled with two disambiguating rewrites, with the rest having three or more.

## A.6 Ambiguity category annotation

The authors construct a taxonomy of ambiguity types by reviewing AMBIENT examples and categorizing possible sources of ambiguity, described in Table 9.

Then two of the authors annotate 100 randomly

Write pairs of sentences that are related to each other in the same way.

Sentence 1: ***In the past,*** I have been of the opinion that a free market economy is a superior economic system.
Sentence 2: I have changed my mind and now believe that a planned economy is superior.
Sentence 1: ***I would like to*** go to the circus.
Sentence 2: I have never been to the circus.
Sentence 1: ***For a long time,*** this concept of "collective responsibility" was more important than the need to protect the individual.
Sentence 2: This concept of "collective responsibility" is no longer important.
Sentence 1: ***When I was young,*** I was obsessed with the supernatural.
Sentence 2: I am not obsessed with the supernatural anymore.
Sentence 1:

Table 8: Prompt template for GPT-3 used to create unlabeled examples for annotation, formatted with an actual set of in-context examples. In-context examples are from WANLI and found automatically via nearest neighbors in `[CLS]` token embedding space of an NLI model finetuned on WANLI. All the examples demonstrate a shared ambiguity pattern where sentences about the past or desires about the future induce a cancellable implicature about the present. For instance, *When I was young, I was obsessed with the supernatural* implies that "I" am no longer obsessed, and *I would like to go to the circus* might be taken to imply (more tenuously) that "I" have not been before.

sampled examples from AMBIENT for the ambiguity type. Each ambiguity is labeled with one category; examples may have multiple categories when they contain multiple ambiguities (e.g., both premise and hypothesis are ambiguous, or one sentence has multiple ambiguous parts). When multiple categories are plausible for a single ambiguity (e.g., a word is *lexically* ambiguous but *pragmatics* encourages the reading of one over the other), we choose the first one in the order of the table (here, *lexical*).

Note that the distribution of ambiguity in AMBIENT does not necessarily reflect that of naturally-occurring ambiguity.

# B  Crowdworker experiment details

## B.1  The crowdworkers

To qualify workers, we designed a qualification test with 5 questions that paid $5.00, open to the 64 annotators who revised and labeled NLI examples for the creation of WANLI. Of the 43 workers taking the test, 34 passed, though only 29 participated

| Category | Description |
|---|---|
| Lexical | A lexical item has different senses |
| Syntactic | Different syntactic parses lead to different interpretations |
| Figurative | Literal and figurative readings are present |
| Pragmatic | Literal and pragmatic interpretations are present |
| Scopal | Ambiguity from the relative scopal order of quantifiers OR the scope of particular modifiers |
| Coreferential | Ambiguous coreference |
| Other | Ambiguity that does not fall into the above categories |

Table 9: Ambiguity categories.

in the actual project. Through a poll taken after the annotation phase was completed, we find that all but one of the participants spoke English as a native language.

For the remainder of the study, crowdworkers were paid $0.40 per NLI example, which involved labeling the original ambiguous example, assessing the plausibility of three interpretations, and finally labeling three (closely related) NLI examples. At the end of data collection, we aggregate the earning and time spent from each crowdworker, and find that the median hourly rate was $19.13.

## B.2  Setup details

To create a "distractor" sentence among the true disambiguations, we use back-translation with Yorùbá by employing the NLLB model (Meta, 2022) with greedy decoding for both Eng→Yor and Yor→Eng.

In case the generated distractor was an exact copy of the original ambiguous sentence, we repeat the Yor→Eng leg of backtranslation with multinomial beam search, with a beam size of 5.0, top $p = 1.0$, and temperature $t = 2.0$. Of the 5 sequences returned, we randomly choose a sequence that is distinct from the original source sentence.

For instance, "*It is currently March, and they plan to have their wedding scheduled for next December*" is back-translated to "*It is March, and they are to be married in December,*" which is a faithful though somewhat lossy paraphrase, and 8/9 crowdworkers consider this a possible interpretation. On the other hand, "*There will be more interesting seminars next quarter*" is back-translated to "*There will be many more exciting conventions in the next half,*" which is not a faithful paraphrase and considered a possible interpretation by 1/9 workers.

# C  LM Experiment details and discussion

## C.1  Generating Disambiguations

For the test in §4.1, there is a different template for when the **premise** is ambiguous and when the

**hypothesis** is ambiguous. For simplicity, we exclude the 6.9% of examples where both the premise and hypothesis are ambiguous. The former template is shown in Table 2; the latter contains only minor modifications. The instruction is "*In each example, you will be given some* **context** *and a* claim. *Unfortunately, the* claim *has some ambiguity that affects whether it is correct. Enumerate two or three interpretations of the* claim *that lead to different judgments about its correctness.*" Then, immediately following the statement of the context and claim, "*We don't know, because the* claim *can be interpreted in many different ways:*".

**EDIT-F1**  The EDIT-F1 metric represents a disambiguation by its added and deleted unigrams, and computes the F1 score between the reference and the prediction. For instance, the ambiguous sentence "*We're afraid that LMs aren't modeling ambiguity*" can be disambiguated with edits ⟨ `-afraid` , `+worried` ⟩. Predicted edits ⟨ `-modeling` , `+representing` ⟩ would receive an EDIT-F1 of zero, whereas sentence-similarity metrics like BLEU would give undue credit for the high overlap between preserved portions of the ambiguous sentence.

**Analysis**  One strategy for attempting disambiguation we observe across model classes is restating the ambiguous sentence with additional context that directly affirms or negates the hypothesis, rather than making a targeted revision to clarify the ambiguity. In some cases, this "shortcut" does lead to technically correct disambiguations (and marked as such in human evaluation). For instance, for

> P: He always ignores his mother's advice to follow his own dreams.
> H: He follows his dreams.

`ChatGPT` disambiguates the premise by restating it, followed by "*and therefore does follow his dreams*" versus "*and therefore does not follow his dreams.*" The former forces the interpretation that he ignores her advice *in order to* follow his dreams; the latter the interpretation that his mother's advice *is* for him to follow his dreams. Thus, the human-judged correctness may overestimate the models' ability to precisely report the source of ambiguity.

## C.2  Recognizing Disambiguations

For the test in §4.2, accuracy on each template is shown in Table 10.

|  | 1 | 2 | 3 | 4 | Avg |
|---|---|---|---|---|---|
| FLAN-T5 (xxl) | 85.9 | 28.2 | 100.0 | 11.6 | 56.4 |
| LLaMa (65B) | 96.1 | 92.1 | 11.8 | 19.9 | 55.0 |
| GPT-3 (davinci) | 46.2 | 69.0 | 45.0 | 71.1 | 57.8 |
| InstructGPT (-003) | 71.9 | 18.1 | 81.0 | 27.5 | 49.6 |
| ChatGPT | 81.5 | 51.7 | 74.5 | 23.4 | 57.7 |
| GPT-4 | 91.6 | 68.8 | 81.8 | 9.9 | 63.0 |

Table 10: Accuracy of LMs on the four templates from the True/False evaluation in §4.2. The **Avg.** column is the one reported in the **T/F Acc.** column of Table 4.

## C.3  Recognizing Interpretation-Specific Continuations

This section includes implementation details and discussion for the test in §4.3.

**KL divergence**  For a given disambiguation $d_i$, let $X$ be a random variable equal to

$$x_c = \log \frac{P(c \mid d_i)}{P(c \mid a)} \quad \text{with prob. } p_c = P(c \mid d_i)$$

In §4.3, we calculate the mean over $X_j$, independent and identically distributed copies of $X$:

$$\bar{X}_n = \frac{1}{N} \sum_{j=1}^{N} X_j$$

First we show that $X$ is an unbiased estimator for the KL divergence.

$$\begin{aligned}
\mathbb{E}[\bar{X}_n] &= \mathbb{E}[X] \\
&= \sum_{c \in \mathcal{X}} p_c x_c \\
&= \sum_{c \in \mathcal{X}} P(c \mid d_i) \log \frac{P(c \mid d_i)}{P(c \mid a)} \\
&= D(P(\cdot \mid d_i) \mid\mid P(\cdot \mid a))
\end{aligned}$$

where the first step follows from the linearity of expectation.

And from the law of large numbers, we observe that $\bar{X}_n$ tends to the KL divergence in the limit.

$$\lim_{n \to \infty} \bar{X}_n = \mathbb{E}[X] = D(P(\cdot \mid d_i) \mid\mid P(\cdot \mid a))$$

**Prepending a stem**  We append one of two stems to the beginning of the disambiguation (or distractor), for both generating continuations and measuring the likelihood of generated continuations. For instruction-tuned models, we append the prompt "`Write a story.↵↵`," so that generating on-topic continuations is consistent with its instruction-following objective. For vanilla LMs, we append

a start quotation mark ", which we find leads to significantly more topical continuations; otherwise, models may generate a newline ↵ and proceed to a new topic.

**Creating the distractor** To create the distractor for an ambiguous sentence, we tokenize the sentence using `spacy` and randomly select a word $w$ with the tag `NOUN` or `PROPN` (proper noun). Then we find the category node $c$ where $w$ has the `IsA` relation to $c$, i.e., $w \rightarrow c$, with the largest weight. Finally, we randomly sample a same-category node $w' \neq w$, representing a single word, such that $w' \rightarrow c$.

Sometimes this replacement is not viable, e.g., when there are no nouns in the sentence, the noun is not in ConceptNet, or there are no same-category words. In this case, we next attempt to replace a pronoun with another heuristically-determined pronoun; failing all else, we randomly replace any noun or pronoun with the word "*corgi*."

**Generating continuations** Given either a true disambiguation or distractor as context, we generate continuations by sampling 100 single-sentence continuations from the full probability distribution, i.e., with top $p = 1.0$. To obtain a single sentence, we stop generation when a sentence-ending punctuation mark (one of !, ?, and .) is generated, and append a period back.

**Limitations** Finally we discuss some limitations we observed with this test. First, the likelihood of a continuation conditioned on context depends not only on the *meaning* of the context, but also *surface-form attributes* like the style and tone, which is a confounding factor in this experiment. Indeed, we observe that there can be a stylistic mismatch between original ambiguous sentence and its disambiguation, often with the latter being more stinted and formal. Generated continuations thus match the formal style, and have lower likelihood under the ambiguous sentence than a semantically equivalent, more casual paraphrase.

In addition, the "closeness" of the distractor affects how easy or challenging the test is. We find that in most cases, the noun replacement procedure creates a sentence which we would expect to have a substantially different set of plausible continuations, potentially leading the test to be too "easy". Yet this varies with the noun being replaced, the replacement chosen, as well as the overall sentence in which it appears. Nonetheless, we require the

distractor for this test in order to make a judgment about the performance of the model.

## D Multilabel Model Experiments

### D.1 Methods

Here we describe the setup of NLI models that predict multiple labels as output (§5.1). **Multilabel models** train separate binary classifier heads for each label on top of the transformer output. During inference, the labels are independently selected based on a threshold (shared across labels) tuned on the development set to maximize F1. **Regression models** train a regressor into $[0, 1]$ that represents the probability of hypothesis being true given the premise. The development set is used to select a mapping from each NLI label into a continuous sub-range, and at inference time we pick all labels whose ranges overlap with the regressed value. **Classifier over sets** is a seven-way classifier over the power set of NLI labels minus the empty set. As it directly predicts a set of labels, this model requires no threshold tuning.

The median thresholds across 5 seeds from our experiments are shown in Table 11.

### D.2 Training Details

For models from prior work, we replicate the training details to the best of our ability. All models are based on `roberta-large`.

The UNLI model (Chen et al., 2020) is trained on SNLI's training set (heuristically mapped to regression labels) for 1 epoch, then trained on *u*-SNLI (human-annotated with regression labels) for 3 epochs.

The AmbiNLI model (Meissner et al., 2021) is first pretrained on single-label data from SNLI + MNLI for 3 epochs, then further finetuned on AmbiNLI for 2 epochs. AmbiNLI examples have distributional outputs, and is sourced from the development set of SNLI and MNLI (which contain 5 labels) and train set of UNLI (which are heuristically mapped to soft labels).

The Distribution Distillation model (Zhou et al., 2022) is trained for 2 epochs on SNLI + MNLI training examples that are re-annotated with the distributional output of a teacher model. The teacher model is a traditional three-way classification model trained on SNLI + MNLI.

Finally, the multilabel model from Jiang and de Marneffe (2022) is trained on the development

| | Model | Thresholds |
|---|---|---|
| **Reg.** | Uncertain NLI (C+20) | E: $(0.69, 1.0)$
N: $(0.01, 1.0)$
C: $(0.03, 0.71)$ |
| **Dist.** | AmbiNLI (M+21)
Dist. Distillation (Z+22) | $-3.43$
$-1.55$ |
| **Class.** | MNLI (M+18)
WANLI (L+22) | $-2.68$
$-1.19$ |
| **Multi.** | Multi-label MNLI (M+18)
Multi-label WANLI
Set classifier on WANLI | $-2.78$
$-1.97$
N/A |

Table 11: Logit thresholds used to map the output of various models to a set of labels, for multilabel prediction experiments (§5). The way these thresholds are obtained and used at inference-time is explained in §D.1.

set of MNLI and ChaosNLI, where a label is considered present if 20% of annotators choose the label. The model with the lowest loss on held-out data over 30 epochs is selected as the final model.

# E Political Claims Case Study

To paraphrase each political claim, we use InstructGPT (text-davinci-003) zero-shot with the simple prompt "*Paraphrase the text.↵↵* {Claim} ↵*Paraphrase:*", and decode with top $p = 0.9$, to encourage both correctness and diversity among generated paraphrases.