# OpenReview forum: "We're Afraid Language Models Aren't Modeling Ambiguity"
_EMNLP/2023/Conference — EMNLP 2023 Main_

### Official Review · Reviewer_LLmw · 2023-08-04

**Typos Grammar Style And Presentation Improvements:** line260-262 full stop missing or misp…
**Soundness:** 4

**Excitement:**

4: Strong: This paper deepens the understanding of some phenomenon or lowers the barriers to an existing research direction.

**Paper Topic And Main Contributions:**

This paper presents a new dataset AMBIENT. Which is a linguist annotated benchmark of 1,645 examples with diverse kinds of ambiguity. The authors make a strong case on why ambiguity matters. A series of tests are presented for pretrained LMs in generating, recognising and disentangling ambiguity. They present a case study on flag political claims where focusing on disambiguation is beneficial.

**Questions For The Authors:**

a) With growing number of available datasets for NLI, how do the authors foresee extending and maintaining this dataset?

**Reasons To Accept:**

As the authors point out, ambiguity is an important consideration in communication, and by extension for LMs. This paper is well written and presented. The dataset presented was carefully collated, with details of various phenomena distributions described. The motivation of this study is further exemplified by the case study on political claims.

**Reasons To Reject:**

An easily rectifiable point:
It would be great if the authors explicitly note the language considered (I assume it's English).  "Always name the language you are working on" (Bender 2011: 18)

Bender, E. M. (2011). On Achieving and Evaluating Language-Independence in NLP. Linguistic Issues in Language Technology, 6. https://doi.org/10.33011/lilt.v6i.1239

**Reproducibility:**

4: Could mostly reproduce the results, but there may be some variation because of sample variance or minor variations in their interpretation of the protocol or method.

**Reviewer Confidence:**

2: Willing to defend my evaluation, but it is fairly likely that I missed some details, didn't understand some central points, or can't be sure about the novelty of the work.

---

> ### Author Rebuttal · Authors · 2023-08-29
>
> Thank you for recognizing that ambiguity is “an important consideration for LMs” and our “careful collat[ion]” of AmbiEnt. We hope the following clarifications address your review!
>
> **Re: the Bender rule — see Limitations**
>
> We currently note in the limitations section that we only study English (L. 595). We will state this explicitly in the main text in the next revision.
>
> **On Extending AmbiEnt**
>
> Our main interest was in studying ambiguity through the format of multi-label NLI, as opposed to overhauling the NLI task itself. So it is not crucial to update all NLI datasets to account for ambiguity; all datasets inevitably contain underspecified/ambiguous examples, but they still have merit.
>
> Since this dataset is important for our own research, we expect active maintenance for some time, and have ideas about extending it, e.g., by finding naturally occurring instances of misunderstanding online or constructing examples at scale from Wikipedia disambiguation pages.

---

### Official Review · Reviewer_CZrE · 2023-08-04

**Soundness:** 5

**Excitement:**

5: Transformative: This paper is likely to change its subfield or computational linguistics broadly. It should be considered for a best paper award. This paper changes the current understanding of some phenomenon, shows a widely held practice to be erroneous in someway, enables a promising direction of research for a (broad or narrow) topic, or creates an exciting new technique.

**Missing References:**

Maybe some classic paper about the theory of ambiguity in computational linguistics? For example:

Semantic ambiguity and perceived ambiguity, by Massimo Poesio

**Paper Topic And Main Contributions:**

The current paper investigates ambiguity. To this end, the authors:
1. built an NLI corpus with lexical, syntactic and pragmatic ambiguity, namely, AmbiEnt.
2. conducted a crowdsourcing experiment to confirm that ambiguity is the major source of disagreements among annotators for NLI.
3. based on AmbiEnt, assessed ChatGPT and GPT-4 and concluded that ambiguity is still challenging for them even though they are fine-tuned on underspecification dn subjectivity NLI corpora.

**Reasons To Accept:**

1. The paper investigates a very interesting and vital issue in language, ambiguity.
2. The corpus is not only useful for the assessment of LLMs but also for the general NLI community or even for linguists.
3. The construction process of the corpus is well-designed making the resulting corpus very much reliable.
4. The assessment experiments are inclusive and the conclusions are backed by sound pieces of evidence.

**Reasons To Reject:**

I don't see major issues in this paper. Here are two minor issues.

The results would be more interesting if they can be grouped with respect to (1) how the data was collected and (2) the type of ambiguity (lex, syn or pragmatic).

It is also vital to discuss what types of ambiguity have been covered and what types have not. This help readers to understand the generalizability of the findings.

**Reproducibility:**

4: Could mostly reproduce the results, but there may be some variation because of sample variance or minor variations in their interpretation of the protocol or method.

**Reviewer Confidence:**

4: Quite sure. I tried to check the important points carefully. It's unlikely, though conceivable, that I missed something that should affect my ratings.

---

> ### Author Rebuttal · Authors · 2023-08-29
>
> Thank you for recognizing the “vital issue” of ambiguity in NLP research, the value of our corpus for “assessment of LLMs… or even for linguists,” and our “sound” experiments!
>
> **Analysis by category**
>
> We only manually labeled 100 AmbiEnt examples with category annotations, and thus we are wary of drawing quantitative conclusions. However, we have qualitatively observed that models are quite strong at dealing with syntactic and coreference ambiguities, but much weaker at pragmatic and scopal ambiguities.
>
> For possible future work, we are considering expanding AmbiEnt along with category annotations, in order to do category-based analysis and study systematic preferences in interpretation!
>
> **Which types of ambiguities are covered, and which are not?**
>
> We hope that our category annotation (shown in Table 1) partially addresses this, by breaking down the types of ambiguity (pragmatic, lexical, syntactic, scopal, coreferential, figurative, other) present in our dataset.
>
> Indeed, there are many types of ambiguity that are rare or missing from our dataset because they do not appear much in our data sources (e.g., slang), or they are hard to capture with our set-up (e.g., sarcasm, which would require more conversational context), or they are rare in English (e.g., morphological ambiguities, briefly discussed in Limitations). We will include a more thorough discussion of this in the Limitations section!

---

### Official Review · Reviewer_Eef3 · 2023-08-10

**Soundness:** 3

**Excitement:**

4: Strong: This paper deepens the understanding of some phenomenon or lowers the barriers to an existing research direction.

**Paper Topic And Main Contributions:**

This this article, the authors focus on capturing ambiguity, by (1) creating a dataset of (premise, hypothesis) pairs with ambiguous language (including NLI labels and unambiguous rewrites), (2) evaluating 6 pretrained language models on proposing possible relevant disambiguations, and recognizing disambiguations with GPT-4 achieving the best results, and (3) evaluating roberta-large-based NLI models that predict one single probability value, a distribution over labels, or a multilabel.


**Questions For The Authors:**

* With the AmbiEnt dataset, the (premise, hypothesis) pairs with single NLI labels are considered unambiguous. Wouldn't it be possible to have some ambiguity with the premise or the hypothesis whose interpretations do not affect the NLP label?
* Even though language is ambiguous and we formulate ambiguous communications, not all interpretations of an ambiguous text are created equal. We have the Grice's cooperative principle and, for most of the examples included in the article/appendix, one of the disambiguations seems more likely than the other, which sometimes feels like a stretch. For instance, why say "Patrick did not manage to leave" to mean "whether or not he tried" (disambiguation 2)? "Patrick did not leave" would be more straightforward and adding the extra "manage to" implies an attempt (disambiguation 1).
* Any insights on how some of the values used for probability thresholds and other settings were selected?
* Isn't the distractor used for candidate interpretations of an ambiguous sentence too different from the actual disambiguations given that it is automatically generated by translation back and forth in a low-resource language when it comes to form? Are these distractors grammatically correct? Do they have correct capitalization?

**Reasons To Accept:**

* AmbiEnt (Ambiguity in Entailment) -- collection of 1,645 examples with lexical, syntactic, and pragmatic ambiguities in 35% of examples
* Manual (expert and naive) and automatic efforts are used to create the dataset with several checks in place, including agreement computations
* Several ambiguity-related experiments are conducted, whose findings are outlined in the paper
* Code and data to be made publicly available

**Reasons To Reject:**

* Some issues related to the dataset-generation process. Please see questions below.

**Reproducibility:**

5: Could easily reproduce the results.

**Reviewer Confidence:**

3: Pretty sure, but there's a chance I missed something. Although I have a good feel for this area in general, I did not carefully check the paper's details, e.g., the math, experimental design, or novelty.

---

> ### Author Rebuttal · Authors · 2023-08-29
>
> Thank you for recognizing the value of our dataset and the insightful questions. We hope the following clarifications address your concerns.
>
> **What about ambiguities that don’t affect the NLI label?**
>
> Indeed, we only consider ambiguities in NLI examples that affect the determination of the label. This is by design. Natural language is rife with ambiguities — consider this very innocuous sentence: “*John and I are going to the bank.*” Does this mean John and I are going together, or possibly separately? Are we presently in the process of going, or do we have plans to go in the future? Are we going to the river bank, or the financial institution? [English Resource Grammar](https://delph-in.github.io/delphin-viz/demo/) actually lists 22 different semantic representations of this sentence. It is really hard to study ambiguities at such a fine-grained level (see our footnote 2), and so we purposefully capture ambiguities that affect NLI labels, to provide a natural setting where ambiguities are contextually relevant. We will make this design choice more clear in the next revision!
>
> Note that the presence of other ambiguities does not affect the validity of our experiments. For  §4.1, we provide both the premise and hypothesis, and ask about relevant ambiguities; in §4.1, all of the True/False constructions are valid even if there are more interpretations than we consider, and likewise for the ranking test in §4.3. For §5, of course we provide the full NLI example for multi-label prediction.
>
> **Not all interpretations are created equal**
>
> We agree that for many examples, there is a more likely interpretation! This is sometimes exactly what makes this problem worth studying: as we emphasize in the political claims case study, speakers can take advantage of this to mislead their listeners into hearing something more than they're actually able to defend. For instance, "*Since I took office, Wisconsin now has the 2nd highest health care ranking in the country*" is a clear violation of the Gricean relevance principle, since in reality, Wisconsin was already 1st or 2nd before he took office! [[cite](https://www.politifact.com/factchecks/2015/aug/05/scott-walker/scott-walker-says-he-took-office-wisconsin-has-ran/)] Yet, that is precisely the speaker's intent, to mislead his listeners without necessarily lying. In these cases, it is important to pick out the less likely interpretation!
>
> For the particular example you pointed out, imagine someone saying "*I did not manage to fix the problem*" to imply they made an effort, when in reality maybe they didn’t.
>
> Besides this, there are many cases where speakers are not necessarily considering the range of possible interpretations from the listener’s perspective, OR “relative likelihoods” between different communicators do not align — and miscommunication occurs.
>
> **Probability thresholds are selected using the AmbiEnt dev set (Appendix D.1)**
>
> This information is currently in Appendix D.1, but we will move it to the main paper for the final version. In short, for each model, we use the AmbiEnt dev set to select threshold(s) that would produce the best F1 score. For instance, the regression NLI model, for each label, we search the space of $[0, 1] \times [0, 1]$ for a pair of endpoints. Then at inference time, we predict all labels whose ranges overlap with the regressed value. The actual thresholds are in Table 11.
>
> **The back-translated distractor has correct form**
>
> All distractors have correct grammar and punctuation, and additionally we were careful to manually ensure they are stylistically similar to the disambiguations. In fact, many of the distractors are faithful paraphrases, and crowdworkers indicate that they are plausible interpretations 46.7% of the time (L. 288), meaning that they are reasonable paraphrases.
>
> Here are some examples of back-translations.
>
> - “The novel has not been banned in many schools.” $\rightarrow$ “Many schools have not banned the book.”
> - “It is difficult to believe that the author of such a masterpiece could have been only twenty-three years old when he wrote it.” $\rightarrow$ “It is hard to believe that the author of this remarkable work was only 33 years old when he wrote it.”
>
> The first is a faithful paraphrase, and the second one should say “23” instead of “33” but is otherwise correct.

---

### Official Review · Reviewer_4NEm · 2023-08-13

**Soundness:** 4

**Excitement:**

4: Strong: This paper deepens the understanding of some phenomenon or lowers the barriers to an existing research direction.

**Missing References:**

In the RW, I would have liked to see a discussion of efforts to overcome this and how these fall short of the task you have proposed here…


Efforts to overcome ambiguity:
* CLAM: Selective Clarification for Ambiguous Questions with Generative Language Models: https://arxiv.org/abs/2212.07769
* Assistance with large language models: https://openreview.net/pdf?id=OE9V81spp6B


Promise for finetuning decoders as a way to overcome ambiguity:
* Task Ambiguity in Humans and Language Models: https://arxiv.org/abs/2212.10711

**Paper Topic And Main Contributions:**

In this paper, the authors propose a novel NLI benchmark dataset of 1,645 ambiguous entailment relations (AmbiEnt). They explore LLMs’ (Flan-T5, LLaMa, GPT-3, ChatGPT, InstructGPT, GPT-4) disambiguation of the entailment relations, finding that LLMs struggle to 1) identify ambiguous constructions and 2) provide disambiguating evidence. Calculating the KL divergence between possible disambiguations for a given construction and unrelated distractor constructions, the authors show that models have room for growth in identifying the correct continuation of ambiguous sentences. The authors finetune NLI models on existing datasets with label variation, finding that these models also struggle to predict the label set for the examples in AmbiEnt. Lastly, they well-motivate their work on ambiguity through demonstrating the use of a multilabel NLI model to detect misleading political claims “in the wild.”

**Questions For The Authors:**

A. Did you try alternate prompting strategies when evaluating the pretrained models (e.g. chain-of-thought or self-critique)? If so, what were they?


B. Why do the evaluation setups for models and crowdworkers differ so significantly? Perhaps I am misunderstanding the human evaluations… I would really appreciate some conceptual clarification/reasoning about the setup here!

**Reasons To Accept:**

* Well-motivated: the authors clearly demonstrate that models have significant room for improvement in ambiguous instances. They also show the value in pursuing research in this domain through demonstrating the presence of real-world ambiguous constructions in political spheres.
* Strong dataset contribution: the authors provide a robust NLI dataset of ambiguous entailments that can be used in future work to measure model’s performance with underspecified information.
* Overall, a really compelling pipeline for studying ambiguity further and leaves a lot of headroom for interesting future work!

**Reasons To Reject:**

Weaknesses
* Lacks a good faith effort to improve model’s performance via prompting: although I understand the need for LLMs to disambiguate these ambiguous inputs, I am unconvinced that with robust prompting, that models would be unable to do this. Judging from the prompts provided in Table 2, the authors fail to utilize modern prompting techniques (chain-of-thought, self critique) which demonstrably improve model performance in a variety of contexts. Furthermore, I personally found the prompts difficult to parse as well. I think a more variable prompting setup is necessary to explore. I couldn’t find any information regarding alternate prompts the authors considered/tested prior to settling on the prompting framework in Table 2.
* Mismatched evaluations between human performance and model performance: the authors argue that even the best model (GPT-4) does not match human performance in generating correct disambiguations. The authors utilize crowdworkers to identify if a given disambiguating construction (either model or directly from the dataset) is viable given the premise for the construction. However:
* 1) the authors do not actually test human performance on this task, instead arguing that the crowdworker’s classification of the model’s generations can be directly compared to their judgment of correctness for AmbiEnt itself. I find this logic to be unconvincing as the crowdworkers themselves were not asked to generate disambiguating evidence – AmbiEnt was carefully curated by linguists to purposefully have viable disambiguating evidence. I would imagine crowdsourced disambiguating generations are not as convincing as those in AmbiEnt.
* 2) in line with (1) the methodology for prompting crowdworkers for baseline measurements in Sections 4.1 and 4.2 greatly diverge from the prompts received by the models. For example, when asked to identify disambiguations of a given ambiguous constructions, the crowdworkers received 2 disambiguating constructions and 1 distractor construction. In contrast, models undertook a more complex labeling task in which they were asked to classify a disambiguating example across 4 different interpretations (Table 3).

Nice-to-haves
* When testing LMs’ ability to model context-specific continuations, the authors exclude ChatGPT and GPT-4 because their API does not enable likelihood calculations. Although I understand this limitation, seeing as these two models performed the best on the previous evaluations, estimating likelihood over repeated trials or some other form of likelihood estimation would have been nice to pursue.
* Prior work has demonstrated the potential of finetuning decoder models in ambiguous contexts. I would be interested to see how a decoder model finetuned on a subset of AmbiEnt performs on the proposed elevations. E.g. Task Ambiguity in Humans and Language Models: https://arxiv.org/abs/2212.10711

**Reproducibility:**

5: Could easily reproduce the results.

**Reviewer Confidence:**

4: Quite sure. I tried to check the important points carefully. It's unlikely, though conceivable, that I missed something that should affect my ratings.

---

> ### Author Rebuttal · Authors · 2023-08-29
>
> Thank you for your insightful questions, and we are grateful that you recognize the “strong dataset contribution,”  “value in pursuing research in this domain,” and “headroom for interesting future work”! We hope the following new experiments and discussion address your concerns.
>
> **Exploration of other prompting methods for disambiguation generation (§4.1)**
>
> Note that the prompt in §4.1 is a bit complex because the specified task requires both disambiguation and entailment judgments. This was necessary for our setup to be valid, because a single sentence considered in isolation may have multiple dimensions of ambiguity, while with the full example there is only one that affects the entailment relation. Adding chain-of-thought to this prompt would only make it more complicated, which was your original concern. However, we tried self-critique by asking GPT-4 to critique its last generation, and then revise it based on its critique. We found that **this actually reduced the F1 score from 18.0 to 10.1,** as the style tends to change dramatically in the revision. For instance, the disambiguation “*I saw a crane (the bird) on my walk. Then the claim is true.*” became “*If the term "crane" is interpreted as a type of bird, then the claim "I saw a bird on my walk" is true.*”
>
> Nonetheless, following the spirit of your question about whether GPT-4 can generate disambiguations when given simple and helpful prompts, we ran new experiments **focusing on the simpler task of disambiguation generation alone**. Note that this is *not comparable* to the original task, which requires the predicted NLI label to be correct, but we hope this gives you a sense of how challenging disambiguation is on its own for GPT-4.
>
> We wrote a new prompt that uses “confidence-boosting” (“*You are an English professor*”) and asks for step-by-step reasoning. The prompt is below (in-context examples are from the dev set):
>
> > You are an English professor. For each ambiguous sentence, analyze the sentence step by step based on your knowledge of the English language, then identify a set of distinct interpretations.
> > 1. **Ambiguous sentence:** Alice and Jon disagreed.
> **Reasoning:** It is possible that Alice and Jon disagreed with each other, or that they both disagreed with something else.
> **Interpretations:**
> a) Alice and Jon disagreed with something.
> b) Alice and Jon disagreed with each other.
> 2. **Ambiguous sentence:** I asked the participant if they were a US citizen or a Green Card holder.
> **Reasoning:** It is possible that the speaker thinks the participant is either a US citizen or a Green Card holder, and wants to know which of the two they are. It is also possible that the speaker wants to know whether the participant is either a citizen or Green Card holder, as opposed to neither.
> **Interpretations:**
> a) I asked the participant if they were a US citizen or if they were a Green Card holder.
> b) I asked the participant if they were either a US citizen or a Green Card holder, or neither.
> 3. **Ambiguous sentence:** It takes a long time to find a room in this building.
> **Reasoning:** This may mean that it takes a long time to find an available room due to the building being crowded, or that the building is hard to navigate and it takes a long time to find a given room.
> **Interpretations:**
> a) It takes a long time to find an available room in this building.
> b) It takes a long time to locate a room in this building.
> 4. **Ambiguous sentence:** Some of the furniture in the store is on sale.
> **Reasoning:** This sentence has a cancellable implicature that some, but not all, of the furniture is on sale.
> **Interpretations:**
> a) Some, but not all, of the furniture in the store is on sale.
> b) Some, and possibly all, of the furniture in the store is on sale.
> 5. **Ambiguous sentence:** {ambiguous_sentence}
> **Reasoning:**
>
> Due to the possibility of many valid disambiguations in this setting, we do human evaluation of 50 examples (we mark generated disambiguations as correct as long as it depicts ANY valid ambiguity; often they are different from the reference). The correctness is 48%, showing that GPT-4 can meaningfully disambiguate in many cases, but there is still huge room for improvement; additionally, it is still important for GPT-4 to disambiguate in context (the original setting).
>
> If you have a particular prompting idea that we have not explored, we are happy to experiment with that!
>
> **Comparability of human and model performance**
>
> Thank you for the very thoughtful question about the comparability of human vs. model performance. We will improve the clarity of this point in the paper based on your feedback.
>
> To be clear, we never claim "human performance in generating correct disambiguations," as you are right that we do not test crowdworkers' ability to directly generate disambiguations. The 89.7% statistic is the correctness of AmbiEnt as judged by crowdworkers. This setup is very different from what we subject pretrained LMs to, so we do not actually refer to it as “human performance” anywhere in §4.
>
> To summarize, we use the 89.7% statistic as a comparison point in two places:
>
> 1. In §4.1, we say that the human evaluation of LM-generated disambiguations "can be directly compared to the crowdworker-judged correctness of AmbiEnt itself at 89.7%" (L. 349 -351). This is true because model-generated disambiguations and AmbiEnt disambiguations are subject to the same evaluation. Here, we are not purporting to make any claim about human performance on the task. Instead, crowdworkers are a fixed evaluation method, and we are providing a way to contextualize the performance of LMs. (That is, 89.7% is a kind of upper bound.)
>
> 2. In §5, where we evaluate multi-label NLI models, we do recontextualize 89.7% as human performance. The idea is that if crowdworkers can recognize disambiguations and identify the corresponding label for each disambiguation, then they would be able to choose the correct set of labels, which is the same task given to multi-label NLI models.
> To be more clear here, we can instead describe 89.7% as a “crowdworker oracle performance," where crowdworkers effectively predict the set of labels by being presented with possible disambiguations.
>
> In any case, to reduce confusion, we will not introduce the 89.7% statistic in §3.2 as “human performance,” and instead describe it as “crowdworker-judged correctness” of the dataset. Please let us know if this would address your concerns.
>
> **Are there ways to obtain sequence probabilities from GPT-3.5/4 for §4.3?**
>
> If the GPT-4 API allowed access to its probabilities, we agree these results would be illuminating for §4.3.
>
> In theory, it is possible to estimate the probability of sequences using Monte Carlo Sampling. However, this is only tractable when the output space is small (e.g., when the probability mass is concentrated on “*Yes*” or “*No*”). With longer sequences, the output space grows exponentially, and one needs intractably many samples to obtain an estimate with low expected error — making it infeasible both in terms of time and API cost.
>
> **Discussion of prior work about overcoming ambiguity**
>
> Thank you for sharing the interesting related work, we will discuss these papers in the next revision. In this work we are more interested in whether LMs possess an ability, rather than whether they can learn it through finetuning (which is a distinct research question). Nonetheless, we did finetune RoBERTa on existing NLI datasets with label variation in §5, in order to connect with the extensive work on predicting disagreement in NLI (which all tune RoBERTa); the best EM accuracy here is 43.6% (Table 5). We acknowledge that finetuning is a promising method for creating dedicated disambiguation models.

---

### Meta-Review · Area_Chair_tcww · 2023-09-18

**Recommendation:** 4

**Metareview:**

This paper presents a new NLI benchmark and experiments using LLMs. The authors study several ambiguities and demonstrate that crowdworkers disagree less after they consider their disambiguations.

All the reviewers are positive.

The dataset is small (~1,500), and that is fine. Yet the authors only calculated agreement with 50 samples (Section 2.4). I am  surprised they did not do it with all the samples. It is also unclear if the agreement were calculated before or after discarding. The resource is the main contribution of the paper (and the idea / problem). I don't think the prompting experiments add a lot of value other than to say "off-the-shelf LLMs cannot solve this problem" (which is expected).

---

### Decision · Program_Chairs · 2023-10-07

**Decision:**

Accept-Main

**Comment:**

This paper presents a new NLI benchmark and experiments using LLMs. The authors study several ambiguities and demonstrate that crowdworkers disagree less after they consider their disambiguations.

All the reviewers are positive.

The dataset is small (~1,500), and that is fine. Yet the authors only calculated agreement with 50 samples (Section 2.4). I am  surprised they did not do it with all the samples. It is also unclear if the agreement were calculated before or after discarding. The resource is the main contribution of the paper (and the idea / problem). I don't think the prompting experiments add a lot of value other than to say "off-the-shelf LLMs cannot solve this problem" (which is expected).